# Surgical Indications and Outcomes of Resection for Pancreatic Neuroendocrine Tumors with Vascular Involvement

**DOI:** 10.3390/cancers14092312

**Published:** 2022-05-06

**Authors:** Amy Y. Li, Brendan C. Visser, Monica M. Dua

**Affiliations:** Division of Surgical Oncology, Stanford University School of Medicine, Stanford, CA 94305, USA; amyyli@stanford.edu (A.Y.L.); bvisser@stanford.edu (B.C.V.)

**Keywords:** pancreatic neuroendocrine tumors, vascular involvement, vascular reconstruction, mesocaval shunt

## Abstract

**Simple Summary:**

Pancreatic neuroendocrine tumors (pNETs) are a heterogenous group of rare epithelial neoplasms. For most patients, surgery remains the only treatment modality to cure pNETs, and is recommended for patients with surgically resectable disease. Many of these tumors are non-functional tumors and do not produce clinical symptoms, so patients may present with locally advanced tumors, which invade surrounding organs or neighboring blood vessels. The presence of vascular involvement had previously been considered a contraindication to surgery, but, in recent years, at centers with considerable experience, aggressive surgery to remove pNETs with vascular reconstruction has been performed safely and with good long-term survival. In this review, we will discuss the considerations for resectability, review novel surgical approaches, and present the available evidence on the immediate and long-term postoperative outcomes.

**Abstract:**

Complete surgical resection of pancreatic neuroendocrine tumors (pNETs) has been suggested as the only potentially curative treatment. A proportion of these tumors will present late during disease progression, and invade or encase surrounding vasculature; therefore, surgical treatment of locally advanced disease remains controversial. The role of surgery with vascular reconstruction in pNETs is not well defined, and there is considerable variability in the use of aggressive surgery for these tumors. Accurate preoperative assessment is critical to evaluate individual considerations, such as anatomical variants, areas and lengths of vessel involvement, proximal and distal targets, and collateralization secondary to the degree of occlusion. Surgical approaches to address pNETs with venous involvement may include thrombectomy, traditional vein reconstruction, a reconstruction-first approach, or mesocaval shunting. Although the amount of literature on pNETs with vascular reconstruction is limited to case reports and small institutional series, the last two decades of studies have demonstrated that aggressive resection of these tumors can be performed safely and with acceptable long-term survival.

## 1. Introduction

Pancreatic neuroendocrine tumors (pNETs) are a rare type of pancreatic neoplasms, only comprising about 7% of all pancreatic tumors, with an incidence of 2–3 per million [1,2]. The treatment of these tumors has always caused controversy, as they are such a heterogeneous group of tumors, including both non-functional and functional tumors (depending on if they secrete hormones) [3], sporadic or inherited (associated with a hereditary syndrome) versions of tumors [4], and different pathologic grades of tumors that lead to varying natural histories, from very indolent with metastatic disease to a more aggressive tumoral process [5]. For most patients, surgical resection remains the only treatment modality to cure pNETs, and should be considered with potentially resectable disease in patients who are medically fit to tolerate surgery. As most of these tumors are non-functional (60–90%), they can often present with very advanced disease at the time of diagnosis, when the mass effect of the tumor bulk finally starts to produce symptoms. Recently, the European Neuroendocrine Tumor Society (ENETS) identified the treatment and prognostic capability of borderline resectable, locally advanced invasive disease of pNETs as an unmet need in the study of these tumors [6].

The concept of vascular reconstruction has already been widely discussed in the treatment of locally advanced pancreatic adenocarcinoma. The 2009 expert consensus statement advocated for pancreatic resections with vein resection/reconstruction as a recommended standard of practice for pancreatic adenocarcinomas locally invading the portal vein (PV), superior mesenteric vein (SMV), or porto-mesenteric confluence in institutions experienced and capable of doing these technically challenging operations [7]. In this group of patients, multiple studies have demonstrated that pancreatectomy with venous reconstruction provides acceptable morbidity, mortality, and better survival rates compared to unresected patients [8,9,10]. The role of surgery with vascular reconstruction in pNETs is less defined, and there is considerable variability in the use of aggressive surgery for these tumors. Up until the time of the ENETS working sessions on the unmet needs of pNETs in 2016, a literature search of only 14 publications had been described as reporting perioperative and/or long-term results of pancreatic resections with vein reconstruction in 37 patients with locally advanced pNET. Because these tumors are rare, there have been no systematic studies on the ability to surgically resect these locally advanced tumors with the involvement of major vascular structures, but the retrospective data available support an aggressive approach to surgical resection in carefully selected patients [11]. The aim of this review is to discuss the role of vascular reconstruction in locally advanced pNET, present the technical considerations in different scenarios of vessel involvement or occlusion, and report the outcomes of these advanced procedures.

## 2. Preoperative Assessment

Patients undergoing a preoperative assessment of their tumor are standardly evaluated by either computed tomography (CT) or magnetic resonance imaging (MRI), given their excellent sensitivity for detecting vascular involvement in patients with pancreatic adenocarcinomas or NETs [12]. The technology for CT has greatly improved over the years, with multi-slice, thin sections, and multi-phasic evaluation of the vascular anatomy. In addition, there is the availability of post-processing with 3D reconstructions and multiplanar reconstructions that allow the surgeon to plan surgery effectively. There are many established radiographic criteria that allow accuracy in diagnosing vascular invasions, such as circumferential involvement of the vein of more than 180 degrees, vascular occlusion with the formation of collateral vessels, a mass effect along the length of a vessel, tear dropping of the vein, indicating tumor tethering, or loss of the fat plane between the tumor and the vessel [10]. Endoscopic ultrasound is another useful tool for direct imaging of the pancreas, and has good accuracy for diagnosis (with fine-needle aspirate), staging, and detection for local venous invasion, but can be largely operator dependent and does not give a large overview or road map when planning targets for vascular reconstruction.

An important observation that has come up in previous reports of resection of pNETs is that, despite the reports of excellent sensitivities of CT imaging for identifying involvement of major vessels, there can also be false-positive results, where the radiologic abutment or invasion is not synonymous with the actual vascular involvement at the time of surgery. In one early report of surgical resection for larger pNETs >4 cm, only 50% of all patients who showed evidence of tumor involvement of the PV or SMV (due to a lack of fat interposition of the tumor and vein) at the preoperative evaluation were confirmed to have true invasion at the time of surgery [13]. In another previous report of 46 patients with preoperative CT evidence of major vessel involvement, out of the 36 patients who were thought to have either PV or SMV involvement, only 25% required vascular reconstruction, despite the mean tumor size of 5 cm. Often, the pNET was found to be abutting or distorting the nearby vessel, without direct invasion. This would indicate that, previously, many patients that may have softer radiographic signs of vascular abutment may often be denied surgery on the basis of vascular involvement, but actually not have true vessel invasiveness. These patients clearly have to be evaluated and treated at a center where vascular control and surgical techniques can be employed in the event of necessary reconstruction, but may represent a situation in which the tumor can be carefully dissected. In more recent years, as the quality and techniques of CT imaging have improved with the clarity of the tumor–vessel interface, we have observed a decrease in the gap between preoperative assessment and intraoperative findings. Another report from 2019 of locally advanced pNETs described 15 patients with SMV involvement preoperatively and 15 patients at the time of surgery who required resection of the SMV/PV confluence [14].

Although infiltration of adjacent vessels is more common overall, another rare finding with pNETs is the presence of tumor thrombus within the portal, splenic, or mesenteric veins. This may be clinically relevant, contributing to portal hypertension with gastric varices or gastrointestinal bleeding. Tumor thrombi may directly extend from the tumor as an appendage into the lumen of adjacent veins, causing enlargement of the affected vessel. In these cases, MRI may offer imaging advantages through the use of multi-phasic flow-sensitive sequences for the detection of tumor thrombi or for characterizing the composition of the thrombus [15]. Contrast enhancement can also differentiate tumor thrombi from bland thrombi, with the former strongly enhancing as pNETs would after intravenous contrast. Tumor thrombus in association with pNETs has also been described in gallium-DOTATATE imaging [16].

## 3. Technical Considerations

There is no established standard for the surgical management of venous involvement in the resection of pNETs, and many strategies have been reported through various case studies. Patients all require high-resolution cross-sectional imaging to evaluate individual considerations, such as anatomical variants, areas and lengths of vessel involvement, proximal and distal targets, and collateralization secondary to the degree of occlusion.

### 3.1. Thrombectomy

Tumor thrombosis is an uncommon finding and may be associated with about 5% of pNETs [17]. It can be treated with the extraction of all thrombi, either through thrombectomy or in combination with vascular resection. For thrombectomy candidates, a few factors should be confirmed with intraoperative ultrasound. The thrombus should be mobile within the vein, well-demarcated circumferentially, and should involve the vessel as more of an appendage in the lumen. Proximal and distal venous control is established, and a longitudinal venotomy on the anterior surface of the vein is made to expose the thrombus. The mobile thrombus is extracted, and the vein is closed with a lateral venorrhaphy or by a venous patch [9,17]. The tumor thrombus can be at the level of the PV or the PV/SMV confluence; often in patients with pNETs, this can be a result of tumor invasion into the splenic vein with extension into the portomesenteric confluence.

### 3.2. Traditional Vein Reconstruction

In a case when a portion of the circumference of the vein is involved, but the lumen remains patent, traditional vein reconstruction can be performed in a variety of ways. For pNETs in the head of the pancreas, complete mobilization of the specimen is accomplished, including uncinate dissection of the superior mesenteric artery (SMA). This would leave the specimen attached at the site of vein involvement only. Proximal and distal control of the PV or SMV is attained with vascular clamps, and the vein is resected en bloc with the tumor. Adequate outflow can typically be preserved throughout the dissection, until the point of clamping and removal of the tumor. Venous reconstruction can be completed via several established techniques. For sidewall adherence, a longitudinal ellipse of the vein can be resected and reconstructed with a transverse venorrhaphy or patch venoplasty, so as not to narrow the vein. For segmental resection of the vein (with or without splenic vein preservation), primary end-to-end closure can be performed for shorter segments (<2.5 cm) with adequate mobilization, to allow tension-free reconstruction (Figure 1). For longer segments of involvement, an interposition graft with the preference of other native autologous vein conduits (primarily the internal jugular (IJ) vein, renal vein, or superficial femoral vein) is another surgical strategy for reconstruction [9,18].

### 3.3. Complete Venous Occlusion with Collaterals

#### 3.3.1. Early Vein Reconstruction for Left-Sided pNETs

Large left-sided tumors with complete PV/SMV occlusion create a surgical challenge, given the presence of extensive venous collaterals (Figure 2). If a traditional approach is taken, where the venous reconstruction is deferred until after the removal of the pancreatic specimen, the dissection will proceed in the midst of significant portal hypertension, with a high risk of intraoperative bleeding. Furthermore, division of the venous collaterals, which is performed using a typical left-to-right approach, compromises outflow and worsens portal hypertension and mesenteric congestion. By approaching these tumors with early in-line vascular reconstruction, portomesenteric flow is restored, and collaterals are decompressed, allowing for safer specimen dissection. The technique for these tumors has been described previously [19]. Early proximal control of the PV and distal control of the SMV and its branches are obtained. This is facilitated by division of the coronary vein (if it enters into the PV), the gastroepiploic vein, and the middle colic vein. When feasible, the splenic artery is also ligated to reduce splenic inflow and congestion of the specimen. The pancreas is divided to expose the involved segment of the PV/SMV confluence. The vein is divided proximal and distal to the margins of portomesenteric occlusion, and that vein segment is lifted laterally to the left with the specimen. An interposition vein graft is sewn in an end-to-end fashion, beginning with the SMV side. After vascular reconstruction, left-sided pancreatectomy and dissection are completed in a right-to-left manner.

#### 3.3.2. Mesocaval Shunting

The surgical strategy of mesocaval shunting involves the creation of a shunt between the SMV and inferior vena cava (IVC) to allow diversion of the portomesenteric blood flow and variceal decompression during pancreatic dissection (Figure 3). Selective preoperative splenic artery embolization may be necessary to reduce portal hypertension when access to the splenic artery intraoperatively would be challenging. The internal jugular vein is the preferred conduit for shunting and reconstruction. The superior mesenteric vein (SMV) is isolated and divided high in the mesentery, and then an end-to-end anastomosis is created with the vein graft. The graft is passed through the mesocolon for the most direct path to the IVC. An end-to-side anastomosis is created between the IJ graft and IVC. After completion of pancreatic dissection and resection, the shunt is taken down from the IVC with a vascular stapler. The graft is trimmed down to size for a tension-free end-to-end anastomosis with the PV. The remainder of the pancreaticoduodenectomy reconstruction proceeds in the usual fashion (Figure 3).

## 4. Reported Outcomes

The growing literature focused on venous resection for locally advanced pNETs is currently limited to observational case series and case reports from the past two decades. The outcome variables examined are varied across publications, but generally include the length of stay, postoperative morbidity, readmissions, patency, and, in some reports, survival. Case reports, primarily from Japan, as well as Turkey and Australia, have reported success in surgically addressing locally advanced pNET with vascular involvement. The majority of these involved portal vein reconstruction [20,21,22,23,24,25], while one involved combined portal vein and celiac axis resection [26], and another case with portal vein/superior mesenteric vein reconstruction [27]. Most of the case reports concluded that reconstruction was feasible, with survival ranging from 9 to 25 months, and up to 55 months at the time of publication [20,21,23,24,26]. The remaining two case reports described mortality at 11 and 15 months postoperatively, due to recurrent distant disease [17,22]. The authors purport that, compared to pancreatic adenocarcinoma, where local vascular invasion can often render the tumor inoperable, pNET prognosis is better, and an aggressive surgical resection with vascular reconstruction of pNET with local invasion is warranted if negative margins can be obtained with vascular resection and reconstruction [20,28].

The earliest case series reporting vascular reconstruction for pNET was published in Sweden by Hellman et al. [13]. Of the 31 patients who underwent resection, 4 required PV/SMV reconstruction in the form of a vein patch, vein graft or synthetic graft. There was no perioperative mortality, but it is unclear from the results of the study what the perioperative outcomes for those specific four patients were. Norton et al. initially reported the results of three patients with SMV reconstruction out of a cohort of nine patients in 2003 [29]. There was no perioperative mortality in these three patients, but there was one patient with postoperative pancreatic fistula. There was 100% graft patency on follow-up imaging. In 2011, Norton et al. followed up with additional results from a larger cohort of 42 patients, including 9 patients with PV/SMV reconstruction and 1 with SMA reconstruction [30]. Similarly, there was no perioperative mortality in the entire cohort, but the authors did not report complications for specific patients with vascular reconstruction. However, while survival was decreased in patients with vascular and/or hepatic disease, the 5-year overall survival was improved compared to those patients who did not undergo resection at all (60% vs. 30–40%) [13,29,30].

There were several additional case series with 2–7 patients each [11,31,32]. These cases all involved portal vein reconstruction. Kleine et al. (n = 3) and Fratini et al. (n = 2) similarly concluded that vascular reconstruction is feasible and may prolong survival, but requires careful patient selection [31,32]. In the series from Haugvik et al. [11], the authors also performed arterial reconstruction in three patients (out of their cohort of seven with resection and vascular reconstruction). The average length of stay was 25 days, and 4 of the 7 patients had postoperative complications, but there were no perioperative deaths. The authors concluded that resection with vascular reconstruction can be performed with acceptable morbidity and mortality, similar to previously reported data. In particular, the authors noted that survival was improved in patients without synchronous liver metastases [11].

Then, in 2020, Titan et al. [14] presented a case series of 99 patients diagnosed with locally advanced PNETs, of which 25 patients were found to have preoperative vascular involvement and 17 patients underwent resection with vascular reconstruction (while the remaining patients were able to undergo resection without vascular resection, i.e., tumor was dissected off the vessel). The authors did not report outcomes specifically for the patients with vascular reconstruction. However, the authors concluded that vascular resection was not associated with an increased risk of tumor recurrence, suggesting that consideration for vascular intervention is warranted in this subset of patients. The disease-free survival in this overall cohort was 60%, similar to the prior studies [14].

Preliminary data from our own single-center experience performing vascular reconstruction for pancreatic tumors with complete portomesenteric occlusion include 15 patients diagnosed with pNET. Of these patients, five patients underwent traditional vascular reconstruction (in the standard order of dissection → resection → venous reconstruction), one with early in-line venous reconstruction [19], two with thrombectomy and seven with intraoperative mesocaval shunting. The median length of occlusion was 5.4 cm. There was no perioperative mortality, and the length of follow-up ranged from 7 to 153 months.

Over time, the literature has shown how vascular resection and reconstruction have been increasingly considered for pNETs. Data, however, remain limited to case reports and case series. As authors have previously noted, case reports and case series represent observational data and level IV evidence, but often spark the development and discussion of new techniques and treatment options [11]. While the results and outcomes are favorable, supporting vascular reconstruction for locally advanced pNET as a feasible technique with acceptable outcomes, more data and future studies are needed to better elucidate the outcomes.

## 5. Summary

These cases can be highly complex, and patients should be evaluated in a multidisciplinary tumor board setting with expertise from various specialties (radiology, pathology, medical oncology, radiation oncology, interventional gastroenterology, interventional radiology, and surgery) for appropriate preoperative planning and postoperative care. The procedures require technical expertise in vascular control and experience with multiple forms of revascularization in the case of unexpected findings or greater vein involvement at the time of surgery. Aggressive surgery has been suggested to be beneficial in patients with pNETs and vascular involvement for several reasons, including (1) improved overall survival compared to those who did not have surgery, (2) improved surgical techniques that have led to safe surgery and acceptable complication rates, (3) amelioration of secondary variceal bleeding consequences of the tumor, and (4) medical therapies alone (such as somatostatin analogues, everolimus, sunitinib, chemotherapy, and peptide radionucleotide therapy (PRRT)) have been reported to show short-term disease stabilization or reduce tumor burden, but, in general, have not been able to downsize extensive local disease with some vascular involvement to make it surgically resectable without vascular reconstruction, as is sometimes observed in other tumors [14,30]. Our preference is that all patients being considered for surgical resection be imaged with multi-phasic thin-slicing CT or MRI to evaluate the level and degree of vascular involvement [30]. The use of positron emission tomography (PET), using somatostatin analogues labeled with positron-emitting gallium 68 dota peptides (^68^Ga-DOTATATE), has been shown to be very accurate for detecting many types of NETs [33]. Specifically for pNETs, ^68^Ga-DOTATATE is beneficial in the overall clinical management of patients, its identification of unknown primary tumors, metastases to the liver, and assessment of global tumor burden in extrahepatic disease [34]. However, it does not provide the anatomical detail required for surgical planning of vascular involvement and is not performed routinely for locally advanced pNETs without distant disease. The recent study mentioned above, by Titan et al., included 99 cases of locally advanced pNETs (of which 25 patients were found to have preoperative vascular involvement), and only 4 patients from the total cohort were treated with neoadjuvant chemotherapy or PRRT [14]. Many other case series of pNETs with vascular involvement also did not receive neoadjuvant therapy prior to surgical resection [11]. However, with the surgical techniques developed to offer more challenging levels of total vascular occlusions and resultant varices, many patients at our institution who are evaluated for a mesocaval shunt and graft procedure often receive neoadjuvant chemotherapy in an attempt to shrink the tumor, and also to give a sense of biological stability over time, prior to embarking on a higher risk procedure for revascularization. The key principles to guide surgical decision making for vascular occlusions include the following: (1) patients without suitable proximal or distal targets for reconstruction are not candidates for surgery; (2) limited occlusions isolated to the PV or SMV often allow for the traditional vascular approach (partial or in-line reconstruction), where adequate outflow is maintained throughout the dissection until the point of removal of the specimen and vascular reconstruction; and (3) tumor occlusion of the portomesenteric confluence and resultant varices will make dissection very difficult in the midst of significant portal hypertension, so early venous intervention is beneficial prior to specimen removal.

## 6. Conclusions

Surgery for pNETs is performed to cure disease, provide relief from hormonally active tumors, or prevent serious life-threatening complications related to the tumor, such as obstruction from the mass effect or bleeding from vascular shunts or varices, as a result of splenic vein occlusion. All of these goals may be met with successful resection of these primary pancreatic tumors, even in the setting of local vascular involvement. Advanced surgical techniques, such as early in-line venous reconstruction or mesocaval shunting, allow for the decompression of mesenteric hypertension for safe pancreatic dissection in the setting of complete venous occlusion. Several studies have shown the safety and feasibility of these radical procedures with acceptable disease-free survival and long-term outcomes. Major vascular involvement in pNETs should not be considered a contraindication to surgery, and favorable results support an aggressive approach in these patients at tertiary high-volume centers.

## Figures and Tables

**Figure 1 cancers-14-02312-f001:**
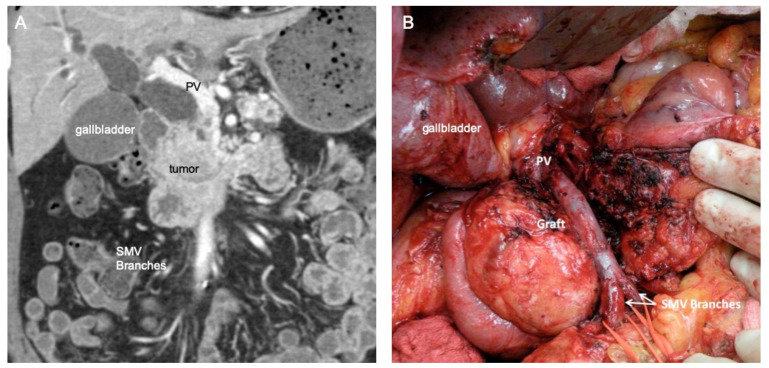
Coronal computed tomography (CT) image of (**A**) a pNET that involves a segment of the superior mesenteric vein down to the level of the jejunal and ileal vein branches, and (**B**) intraoperative photograph of the completed vein reconstruction where the portal vein was mobilized, the inferior mesenteric vein was taken, and distal SMV branches were connected together to facilitate an interposition graft with the internal jugular vein. PV, portal vein; SMV, superior mesenteric vein.

**Figure 2 cancers-14-02312-f002:**
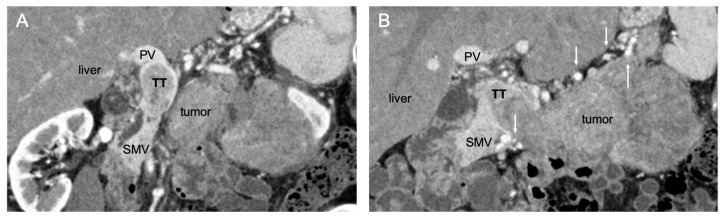
Coronal computed tomography images of (**A**) a large left-sided pNET with complete occlusion of the PV and (**B**) extension of the tumor thrombus and resultant surrounding varices (white arrows). PV, portal vein; TT, tumor thrombus; SMV, superior mesenteric vein.

**Figure 3 cancers-14-02312-f003:**
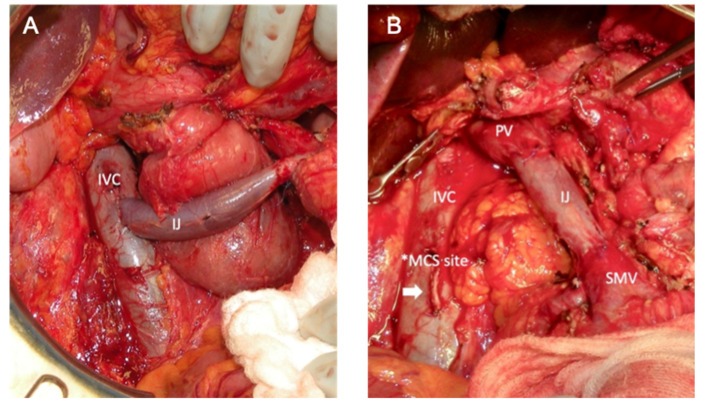
Intraoperative photos of the creation of a mesocaval shunt (MCS) via use of the removed internal jugular vein (IJ). The shunt is between the inferior vena cava (IVC) and superior mesenteric vein (SMV) (**A**) to facilitate dissection of the tumor, and the completed vein reconstruction is shown in (**B**) from the portal vein (PV) to the SMV, with the old MCS site to the IVC shown as a staple line to be able to swing the IJ vein towards the SMV for the distal anastomosis.

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
