# Peer review of "Surgical Indications and Outcomes of Resection for Pancreatic Neuroendocrine Tumors with Vascular Involvement"

_cancers, 2022, doi:10.3390/cancers14092312_

Round 1
Reviewer 1 Report
As the Autors mentioned surgical therapy of NETs with vasucular resection offers a curative approach. This type of surgery is complex and requires a specialized, multidisciplinary setting, and should performed at high-volume centers! This is in my opinion very important to highlight!
The impact of the resection margin status on oncological outcome would be interesting, although data is rare.
Figure 1 and 2 would benefit of a detailed illustration as in Figure 3
Author Response
1. As the authors mentioned surgical therapy of NETs with vasucular resection offers a curative approach. This type of surgery is complex and requires a specialized, multidisciplinary setting, and should performed at high-volume centers! This is in my opinion very important to highlight!
This is very important to highlight and we have included it in a new summary paragraph.
2. The impact of the resection margin status on oncological outcome would be interesting, although data is rare.
The literature on pNET resection with vascular resection has been reported in small series or case reports. A recent lit review on pNET resection plus vein recon focused on 13 other cited reports plus their own series for a total of 14, however only one of these 14 mentioned margin status and this was in a case report. Our institution had a more recent (2020) report of locally advanced pNETs of which 16% had a positive margin but in addition to vascular reconstruction, this series comprised patients with multivisceral resections of stomach, small bowel, stomach, kidney, and adrenals and therefore, the margin status was not stratified based on vascular resection alone. Therefore, the reviewer is correct that the data is extremely rare and was not sufficient enough to include in the review.
3. Figure 1 and 2 would benefit of a detailed illustration as in Figure 3.
These figures have been modified per recommendation.
Reviewer 2 Report
Many thanks to the authors for the efforts in writing this review. The article addresses a complex issue about surgical approach in patients with pNETs and vascular invasion. I think more citations should be added to the text in the first paragraph about epidemiology and subtype of pancreatic NETs.
The article offers a very technical vision on aspects of surgery in pNETs, but data beyond surgical aspects is not mentioned, for example if patients received some peri o pre-operative systemic treatment or co-morbidities of patients included. No data about use of metabolic studies (FDG PET or Ga 68 DOTA) for staging purposes before surgery. I assume these patients are well-differentiated NE tumors but this should be clarified in the paper.
Finally, another important point that can make reading this paper more attractive would be to add some algorithm or recommendations regarding stratification prior to surgery and criteria or pitfalls to considering based on the data presented in the current review.
Author Response
1.Many thanks to the authors for the efforts in writing this review. The article addresses a complex issue about surgical approach in patients with pNETs and vascular invasion. I think more citations should be added to the text in the first paragraph about epidemiology and subtype of pancreatic NETs.
These citations have been added.
2.The article offers a very technical vision on aspects of surgery in pNETs, but data beyond surgical aspects is not mentioned, for example if patients received some peri o pre-operative systemic treatment or co-morbidities of patients included. No data about use of metabolic studies (FDG PET or Ga 68 DOTA) for staging purposes before surgery. I assume these patients are well-differentiated NE tumors but this should be clarified in the paper.
This article was an invited review to focus on the surgical techniques of resection but we have included the utility of DOTA and neoadjuvant chemotherapy in a new summary section.
3.Finally, another important point that can make reading this paper more attractive would be to add some algorithm or recommendations regarding stratification prior to surgery and criteria or pitfalls to considering based on the data presented in the current review.
Thank you, guiding principles for management of types of vascular occlusions have been added to the summary section.

Reviewer 3 Report
Very interesting and enjoyable work to read.
I only have few minor revisions to propose :
- Figure 2 : the figure should be better explained: setting of arrows?
- It's a bit of a shame to show a photo of adenocarcinomas in an article on NETs (even if the iconography is of very good quality)
- Space problem line 241
Author Response
1. Figure 2 : the figure should be better explained: setting of arrows?
Thank you for this suggestion, this was also noted by another reviewer so we added labels to the figure as we had in Figure 3.
2. It's a bit of a shame to show a photo of adenocarcinomas in an article on NETs (even if the iconography is of very good quality)
You are totally right, somehow I did not catch that. I found a different figure that represents pNET vascular reconstruction, thank you.
3. Space problem line 241
This is fixed, thank you.